# A comparison of dynamic balance performance between non-dancers and amateur dancers across three distinct dance genres: A cross-sectional study

Ningyi Zhang[1,2], Sebastián Gómez-Lozano[2], Ross Armstrong[3], Hui Liu[4], Ce Guo[1]*, Alfonso Vargas-Macías[5]

1 School of Athletic Performance, Shanghai University of Sport, Shanghai, China, 2 Performing Arts Research Group-Faculty of Sport, San Antonio Catholic University, Murcia, Spain, 3 Institute of Health, University of Cumbria, Carlisle, England, 4 Biomechanics Laboratory, Beijing Sport University, Beijing, China, 5 Telethusa Centre for Flamenco Research, Cádiz, Spain

* guoce@sus.edu.cn

## Abstract

There is controversy regarding whether dance training improves balance performance. This study aims to compare the dynamic balance performance of non-dancers and amateur dancers, as well as to examine whether differences in dynamic balance exist across various dance genres. Eighty-one participants, including 24 Flamenco dancers (FLA), 15 Latin dancers (LAT), 20 Chinese folk dancers (CHF), and 22 non-dancers as the control group (CG), completed the Y-Balance test. Anterior (YBant), posteromedial (YBpm), posterolateral (YBpl), and composite (YBcom) scores were calculated. All significant differences between groups were observed in YBcom, YBpl, and YBpm ($p < 0.05$), with no differences in YBant ($p > 0.05$). The CHF and FLA achieved higher scores compared to CG in both dominant leg (DL) and non-dominant leg (NDL), and to those of LAT in NDL for YBpm. The LAT scored slightly higher than CG ($p < 0.05$, $d < 0.2$), but it was not statistically significant. Dancers have better dynamic balance except for the YBant direction. FLA and CHF may be more effective in improving dynamic balance compared to LAT. This study was designed by considering previously contentious research findings that may be due to the distinct movement patterns and training methods of professional dancers, as well as the variations across different dance styles. The comparison of balance performance between amateur dancers, rather than professional dancers, and non-participants has significant practical implications for determining whether dance can improve an individual's balance.

**Data availability statement:** All relevant data are within the paper and its Supporting Information files.

**Funding:** This study was funded by Shanghai Key Lab of Human Performance (Grant No. 11DZ2261100 to CG).

**Competing interests:** The authors have declared that no competing interests exist.

## 1. Introduction

Balance performance is defined as the capacity to sustain posture through the interactions of the musculoskeletal and nervous systems, while achieving equilibrium by keeping the center of gravity within the base of support [1] Maintaining an upright posture is a complex process that involves coordinating multiple muscles and joints, while simultaneously integrating feedback from various sensory systems, including the vestibular, visual, and tactile systems [2]. From a dynamic perspective, balance is defined as the capacity to make adjustments that preserve posture during movement [3]. The ability to maintain balance relies on somatosensory, visual, and vestibular sensory systems [4–6]. Balance is a fundamental component of daily functional activities [1] and plays a vital role in injury prevention and improved dynamic balance enhances coordination and stability leading to a decreased risk of falls and injuries [7,8].

Multiple factors influence balance performance, including genetics, age, strength, flexibility, coordination, the condition of the vestibular system, the base of support, and the positioning of the center of mass can affect this ability [9,10]. Furthermore, participation in motor activities and training status [10] can influence balance performance. Previous research indicated that training interventions aimed at improving dynamic balance can lead to significant reductions in fall risk among older adults [11,12]. Thus, fostering dynamic balance is essential for enhancing quality of life and maintaining overall health across various populations and dance participation could potentially contribute to this.

Dance training can significantly enhance physical health across various domains [13,14], including aerobic capacity, cardiovascular function [15], lower body muscular fitness [16–18], flexibility, reaction time, agility, and gait [19]. Moreover, the impact of dance training on dynamic balance performance has been explored, with specific forms such as Thai classical dance [20], Chinese square dancing, ballet, and Latin dance [17,21–23] demonstrating better balance performance than non-dancers [24].

The Star Excursion Balance Test (SEBT) has been utilized to assess injury risks in dancers and athletes [25]. The Y-balance test (YBT), a composite modified version of the SEBT, is a widely used and reliable screening tool for injury risk assessment, focusing on abnormal movement patterns, asymmetry, and dynamic balance [26]. It assesses three directions (anterior, posteromedial, and posterolateral) and measures unilateral balance and neuromuscular control, both of which are critical for many sports [27]. The YBT has been used to assess the relationship between dynamic balance and injury risk [28]. Thus, it is pertinent to evaluate whether this functional test can effectively identify injury risk in high-risk populations. Our study investigated dynamic balance performance via the YBT of non-dancers and amateur dancers from three distinct dance genres namely Flamenco dancers (FLA), Latin dancers (LAT) and Chinese folk dancers (CHF).

However, research findings regarding the impact of dance on balance performance is not without controversy. Some evidences were found in Latin dance, flamenco dance and folk dance. The influence of Latin dance on balance performance remains a subject of debate that some studies demonstrated Latin dancers had better

dynamic balance performance compared to non-dancers [17,23], in contrast, Bojanowska et al. [29] reported that the level of dynamic and static balances of Latin dancers was similar to non-dancers. For flamenco dance, one study separated fifty-two sedentary postmenopausal women randomly to a dance training group or self-care treatment advice group, and indicated a two-month intervention of flamenco and sevillanas training was effective in improving balance in sedentary postmenopausal women [30], while Zhang et al. investigated flamenco dancers balance performance and reported that the score of the Y-Balance test (YBT) of amateur dancers was higher than professional dancers [31]. Regarding folk dance, a study of professional folk dancers and non-dancers indicated that dancers had poorer dynamic balance than the non-dancers [32]. Furthermore, a systematic review indicated when assessed through basic balance tests, dancers exhibited poorer balance compared to untrained individuals [33]. For example, one study investigated balance for competitive dancers and control group for a single task (quiet stance) and a dual task (with a concurrent mental task) on a force plate, and indicated that the postural control of dancers and non-dancers appears to be similar [34].

This may be due to some studies comparing a single type of dance training with non-dancers or testing the balance performance of non-dancers versus professional dancers. Different dance genres employ varying training methods, and professional dancers may possess specific movement patterns that result in varied performances in standardized movement tests [31]. Harmon et al., investigated balance and motor control in dancers and non-dancers with different foot positions and found superior balance and motor control in dancers may be limited to more dance-specific foot positions [35]. Another study indicated the dancers may have greater balance than non-dancers in some but not all tests [36]. Although the current study does not constitute a systematic review, we have observed that the controversy regarding the effect of dance training on balance performance is particularly reflected in the divergent outcomes derived from different research designs. Cross-sectional studies comparing professional dancers and non-dancers often yield inconsistent conclusions, with some reporting superior balance in dancers and others indicating no significant differences or even inferior performance among dancers. In contrast, intervention studies implementing dance training among non-dancers mostly demonstrate significant improvements in balance ability. Therefore, by comparing amateur dancers with non-dancers, this study aims to directly evaluate the actual benefits of dance training on balance performance in the general population, thereby providing evidence to support dance as a feasible health-promoting activity. We hypothesize that amateur dancers will have better dynamic balance performance compared to non-dancers, and the extent of the difference may depend on the specific dance genre in which the amateur dancers participate.

## 2. Methodology

### 2.1. Participants

Eighty-one female participants (mean age: 28.49±6.30 years), volunteered for this study (24 FLA, 15 LAT, 20 CHF, and 22 controls). The control group (CG) did not participate in any regular sport training and dancing. Descriptive characteristics of the participants are presented in Table 1. Participants for FLA, LAT, and CHF group were amateur dancers who engaged in dance for recreational purposes only and attended dance training at least 2 hours per week and had a

**Table 1. Descriptive characteristics of participants (n=81).**

| Characteristics | LAT (n=15) | CHF (n=20) | FLA (n=24) | CG (n=22) |
|---|---|---|---|---|
| Age (years) | 29.80±6.28 | 28.50±6.19 | 29.42±6.61 | 26.59±6.03 |
| Height (m) | 1.68±0.09 | 1.65±0.04 | 1.64±0.05 | 1.67±0.07 |
| Mass (kg) | 58.93±8.51 | 56.85±4.51 | 55.73±4.52 | 59.36±7.82 |
| BMI (kg/m²) | 20.76±2.04 | 20.99±1.56 | 20.64±1.51 | 21.32±2.68 |

LAT: Latin dance group; CHF: Chinese folk dance group; FLA: Flamenco dance group; CG: Control group; BMI: Body mass index.

minimum of 1-year dance experience and had no other type of physical training in the last 3 years. The inclusion criteria for the CG were that participants had no dance experience and had not participated in any type of sport regularly in the previous 3 years. All participants were over 18 years of age and had had no musculoskeletal injuries in the 6 months preceding the test. Participants provided informed consent in writing before the commencement of the study. A stadiometer (Ruhe, China) with used to measure height (cm), a weighing scale (Xiaonmi, China) was used to measure mass (kg). Body mass index (BMI) was calculated as [mass (kg)/height (m2)]. The recruitment of participants for this study was conducted between March 22, 2022, and October 22, 2022. The procedures, associated risks, and potential benefits of the test were comprehensively outlined to the participants prior to their involvement. Prior to participation, written informed consent was obtained from all participants, who provided their signatures on the consent forms. Ethical approval was granted by the Sports Science Experiment Ethics Committee of Beijing Sport University (2022037H), and the study was completed in accordance with the Declaration of Helsinki.

## 2.2. Protocol

All participants were informed about the experimental methods and procedures, and the test was demonstrated by a laboratory technician with 5 years' experience and training in the use of YBT. The test was performed without shoes and prior to performing the test, the participants´ legs length was measured in a supine lying position from the anterior superior iliac spine to the most distal aspect of the medial malleolus using a measuring tape (Y-Balance Test Kit, FMS, USA) [37]. Prior to formal testing, participants performed two test trials in each of the three reach directions. Participants were instructed to stand with the dominant leg (DL) on the center footplate with the great toe at the starting line and with their hands on their hips. Then they were instructed to push the reach indicator in the red target area with the free limb in anterior, posteromedial and posterolateral directions as far as possible while maintaining a single leg stance. After each trial, participants had to return to the starting position. After finishing the dominant leg test, participants performed the same test with the non-dominant leg (NDL). The dominant leg was determined as the leg most often used by the participant to kick a ball [38]. Each participant performed three trials (each trial with three directions) using each leg, and participants were allowed adequate rest time during each test trial. Scores for each direction, anterior (YBant), posteromedial (YBpm), and posterolateral (YBpl) were calculated by dividing the mean reach distance by the participant's leg length [37] and multiplying by 100 to determine the percentage of the leg length [36,39,40]. The composite score (YBcom) was the sum of the three mean reach distances divided by three times limb length, and then multiplied by 100 [41].

## 2.3. Statistical analysis

All data were analyzed using a statistical software package (SPSS IBM Statistics V21.0, IBM, Armonk, New York, USA) with descriptive statistics presented as mean±standard deviation. The normality of the distribution of variables and homogeneity of variance were assessed using the Kolmogorov–Smirnov test and Levene's statistic. Independent sample t-test and Mann-Whitney U tests were used to analyses difference between groups. To explore the bilateral asymmetry (dominant leg vs. non-dominant leg), a paired sample t-test was performed. Differences in groups and dominant leg and non-dominant leg were quantified using a two-way ANOVA. Bonferroni correction factors were used for a post-hoc comparison, to determine where any significant differences occurred. 95% Confidence Intervals (CIs) are presented for significant findings, along with Cohen's d effect sizes (small, 0.20–0.49; moderate, 0.50–0.79; large > 0.80) [42]. Differences were deemed statistically significant at the $p < 0.05$ level.

## 3. Results

The Y-Balance test score of both dominant and non-dominant leg in LAT, CHF, FLA and CG groups is presented in Table 2.

Table 2. The Y-balance test scores of both dominant and non-dominant leg in LAT, CHF, FLA and CG groups.

| | LAT (n = 15) | | CHF (n = 20) | | FLA (n = 24) | | CG (n = 22) | |
|---|---|---|---|---|---|---|---|---|
| | DL | NDL | DL | NDL | DL | NDL | DL | NDL |
| YBcom | 87.91 ± 7.98 | 84.61 ± 6.45 | 87.75 ± 6.57* | 88.59 ± 6.44* | 91.51 ± 6.57* | 90.79 ± 8.03* | 79.84 ± 8.48 | 78.29 ± 8.99 |
| YBant | 71.68 ± 7.58 | 69.86 ± 6.61 | 70.00 ± 4.34 | 70.97 ± 4.83 | 70.69 ± 6.73 | 70.89 ± 6.84 | 66.45 ± 6.90 | 66.21 ± 8.29 |
| YBpl | 93.51 ± 11.93 | 90.72 ± 8.30 | 92.79 ± 9.66 | 92.37 ± 10.60* | 99.92 ± 11.81* | 98.07 ± 13.33* | 84.26 ± 14.32 | 79.77 ± 13.91 |
| YBpm | 97.88 ± 7.93 | 93.57 ± 7.38 | 100.45 ± 9.56* | 102.43 ± 7.39*# | 103.93 ± 9.25* | 103.40 ± 10.27*# | 88.82 ± 8.59 | 88.89 ± 9.66 |

*Significant differences existed in comparison to CG (p < 0.05, and d > 0.2); #Significant differences existed in comparison to LAT (p < 0.05, and d > 0.2). YBcom: Y-Balance test composite scores; YBant: Y-Balance test anterior scores; YBpm: Y-Balance test posteromedial scores; YBpl: Y-Balance test posterolateral scores; LAT: Latin dance group; CHF: the Chinese folk dance group; FLA: Flamenco dance group; CG: Control group; DL: dominant leg; NDL: Non-dominant leg

For the YBcom, the score of CHF (CI: 84.41–91.08; d = 0.17; p < 0.05) and FLA (CI: 88.47–94.55; d = 0.63; p < 0.05) group were greater than the CG in DL (CI: 76.66–83.02). The CHF (CI: 85.26–91.92; d = 0.39; p < 0.05) and FLA (CI: 87.75–93.83; d = 0.71; p < 0.05) group scores were significantly greater than the CG in NDL (CI: 75.11–81.47). There was no significant difference between LAT and CG in NDL (p > 0.05), although the p value was smaller than 0.05 in DL, the Cohen's d effect size was 0.09 which was smaller than 0.2, therefore demonstrating no significant main effect differences.

For the score of the YBpl, the FLA (CI: 95.04–104.80; d = 1.07; p < 0.05) group scores were significantly greater than the CG (CI: 79.17–89.35) in the DL. The CHF (CI: 87.03–97.72; d = 0.52; p < 0.05) and FLA (CI: 93.20–102.95; d = 1.02; p < 0.05) group scores were significantly greater than the CG (CI: 74.68–84.87) in NDL. There was no significant difference between LAT and CG in the DL (p > 0.05) or NDL (d = 0.07; p < 0.05).

For YBpm, the CHF (CI: 96.50–104.40; d = 0.64; p < 0.05) and FLA groups scores (CI: 100.32–107.53; d = 0.24; p < 0.05) were significantly greater than the CG (CI: 85.05–92.58) in the DL. The CHF (CI: 98.48–106.38; d = 0.85; p < 0.05) and FLA group scores (CI: 99.79–107.01; d = 0.97; p < 0.05) were significantly greater than the CG (CI: 85.13–92.66) in the NDL and these two groups scores, CHF (d = 0.36; p < 0.05) and FLA (d = 0.52; p < 0.05) were also significantly greater than LAT (CI: 89.00–98.13). There was no significant difference between LAT and CG in the DL (d = 0.08; p < 0.05) or NDL (p > 0.05).

There was no significant difference between groups for YBant scores or between DL and NDL for all groups in any YBT score (p > 0.05).

## 4. Discussion

The research literature regarding whether dance training improves balance performance is conflicting, which may be due in part to varying methodologies such as the comparison of non-dancers versus professional dancers and different balance tests such as the YBT, biodex stability system [35], balance error scoring system [39], modified bass test of dynamic balance [36], foam and dome test [5], and the star excursion balance test [43,44]. Different dance genres employ varying training methods, and professional dancers may possess specific movement patterns that are different to amateurs therefore resulting in varied performances in standardized movement tests. To account for this, our study investigated non-dancers and amateur dancers from three distinct dance genres and did not use professional dancers.

The FLA group score of YBcom, YBpl, and YBpm achieved greater scores compared to the CG in both DL and NDL. Regarding flamenco dance footwork techniques, there is a high physical demand for dancers [45–47]. Flamenco dance techniques involve a variety of technical steps in which dancers utilize different parts of their feet, including heels and toes, to strike the floor and produce a series of rhythms and loud sounds. Throughout these movements, dancers quickly alternate between different feet and require good balance to maintain stability in the upper body and torso [31,48,49]. In a previous study investigating the impact of balance performance on external load in dancers performing the Zap-3 step, it

was noted that amateur dancers demonstrated greater balance performance improvement in the management of external load in the directions of YBcom and YBpm, in comparison to professional dancers in the YBT [31], therefore FLA training may improve balance performance of individuals who are non-professional dancers.

The CHF group score of YBcom, YBpl, and YBpm was greater than the CG in NDL, and the scores of YBcom and YBpm were higher than CG in the DL also. The impact of Chinese folk dance or fitness dance/square dance incorporating elements of Chinese folk dance on balance performance has also been confirmed; however, most studies focus on the effects on middle-aged and older adults. A previous study conducted a 13-week intervention using a type of Chinese folk dance (Guozhuang dance) and found a significant impact on balance performance. Feng et al. reported that long-term regular participation in Chinese fitness dancing can serve as a preventive strategy to enhance balance and reduce the risk of falls [16]. A systematic review which studied the effect of square dance interventions on the physical and mental health of Chinese older adults noted that several studies have demonstrated the impact of dance on balance performance and utilised tests such as the 8-foot up and go test, unipedal stance with eyes closed and/or open, and the swing feet test [50].

Although the LAT score of YBcom, YBpm in DL and YBpl in NDL were slightly higher than CG, this difference was not statistically significant. Similar findings were reported by Bojanowska et al., [29]who examined the balance performance of Latin American dancers compared a control group, of university students (21–35 years) and found no significant differences in the YBT between groups, and a slightly higher score was observed in YBcom, YBpl, and YBpm for the LAT group, with no differences observed in the YBant direction. A previous study using a 12-week intervention demonstrated the positive impact of Latin dance training on the balance performance of college students [51], Other research groups also reported that LAT training had a positive impact on the balance performance of the elderly [23,52]. A further study also noted that a 12-week intervention of Latin dance and Pilates could improve balance performance in older adults [53,54].

The contrasting results compared to our study may be attributed to several factors. Firstly, the methods used to assess dynamic balance performance vary; our study utilized the YBT, while others may have employed the Flamingo Balance Test-FDT [51], Balance Check 636 [23], Postural Sway task [54], one-leg stance tests [17]. Secondly, the characteristics of the participants differ; some studies utilize college students and older adults [53], whereas our study investigated adult women. Thirdly, some studies are of randomized controlled trial design, while our research is a cross-sectional design. Finally, existing literature presents varying definitions of Latin dance. Latin dance can refer to a category within sport dancing (competitions consist of the Cha-cha-cha, Rumba, Samba, Paso Doble, and Jive) [23] or encompass various styles such as salsa and bachata [51], or mixed training of the tango, rumba, bachata, and basic steps of the social-American system as the LAT group [54]. Therefore, different styles may involve distinct training approaches. Our study used LAT participants with Latin dance of sport dancing, salsa or bachata experience.

The results indicated that the scores of CHF and FLA were higher than those of LAT in the NDL for YBpm, but this difference did not appear in the dominant leg (DL). Although the reason for this result is not yet clear, some studies on dancer asymmetry might provide insights, and future research could explore this further. For example, Prus and Zaletel explored body asymmetries across four different dance genres: standard and Latin American dance, acrobatic rock and roll, breakdance, and hip-hop dance [55]. The findings indicated that dancers in the standard and Latin American styles are the most prone to developing body asymmetries, primarily due to the closed positions characteristic of these dances and their inherent postural asymmetry [55]. However, it is worth noting that this study grouped standard and Latin American dance together for analysis. While both styles belong to sport dancing and require partner coordination and feature asymmetrical closed positions, the authors suggest that differences may exist in movement patterns, physical demands, and training methods. Therefore, this aspect warrants consideration.

All significant differences between dancers and non-dancers were observed in YBcom, YBpl, and YBpm directions, with no differences found in YBant. Similar results were obtained in other studies involving Urpin folk dance, LAT, modern dance, and collegiate dance majors. When compared to the dancers from the Urpin folk dance group and the CG,

analysis revealed no significant difference between the two groups in the YBant. However, differences were observed in the YBpl and YBpm directions [32]. Another study compared LAT (salsa and bachata) with a CG, finding that no differences were noted in YBant, but a slightly better performance in YBcom, YBpl, and YBpm was observed for the LAT group [29]. Researchers also indicated that modern dancers had greater SEBT reach distances in the medial and postero-medial, but not in the antero-medial direction [36].

The lack of superior performance in YBant among the dance group may potentially be attributed to the varying training methods associated with different dance genres. For instance, Indian classical dancers performed significantly better SEBT score in all the three directions of anterior, posteromedial, and posterolateral [56]. Another study found 8-weeks Zumba (salsa and aerobics) training could improve the balance in all eight reaching distances of the SEBT in female college students [57]. In standard dance which is a style of ballroom dancing that includes the waltz, tango, Viennese waltz, slow foxtrot and quickstep, when executing a forward step, dancers typically maintain an upright posture, with one leg acting as the dominant leg, bending at the knee to provide support and balance, while the other leg remains extended forward. Throughout this movement, dancers need to transfer their weight from the back leg to the front leg to maintain stability. Additionally, it is essential for dancers to uphold good posture, keeping the upper body upright and arms positioned in a standard dance frame. In such a training model, significant differences in YBant might be expected compared to the control group. In a previous study, researchers studied the efficacy of a 10-week neuromuscular training of elite youth competitive ballroom dancers and used the YBT [58], which demonstrated a higher score in YBant than our findings. However, our study did not include standard dancers, which represents a limitation of our research. Moreover, when comparing dance with other sports training, dancers' superior posture control is also evident across different directions, including the anteroposterior axis. A recent article by Armstrong [59] studied postural stability scores in female rugby and netball players, as well as dancers, and examined differences in performance between the non-dominant and dominant limbs within each group using the Biodex Stability System. The study found that dancers exhibit superior postural control in the anteroposterior, mediolateral, and overall stability index.

Regarding other dance genres, previous research has demonstrated that ballet dancers achieve higher balance scores compared to other active non-dance athletes [21], In numerical terms, the results of this study indicate that the YBcom for ballet dancers (mean: 105.0±8.2) is higher than that of all groups in our research. The YBant value is similar to our findings at 71.5±4.8, and the YBpm (112.8±9.4) and YBpl (115.5±8.6) scores are elevated above all groups in our study. Thai classical dance may also positively affect balance, as a study comparing the balance performance of Thai classical dancers and non-dancers demonstrated that dancers exhibited superior balance performance compared to non-dancers, utilizing the modified Sensory Organization Test (mSOT) as the assessment method [20]. While Thai Khon Masked Dancers have greater SEBT performance versus non-dancers [60].

## Limitation and future perspectives

While we observed significant associations between amateur dance participation and superior dynamic balance performance, we are unable to establish a causal relationship due to the cross-sectional nature of the study. To address this limitation and provide more robust evidence, future longitudinal studies tracking balance ability before and after individuals engage in dance training. Furthermore, this study focused exclusively on amateur dancers, and the LAT group had the lowest sample size, which may limit the power to detect differences. While this design choice strengthens the inference about recreational dance participation, it also means our findings cannot be directly compared to or explain the discrepant results often observed in studies of professional dancers. While the YBT is a widely accepted and reliable tool for assessing dynamic balance, it primarily measures reach distance rather than providing a direct evaluation of center of mass control or reactive balance responses. Future studies should consider complementing YBT with additional assessments, such as force plate analysis or reactive balance protocols, to provide a more comprehensive evaluation of balance performance.

## Conclusion

Dance training had performance benefits for all YBT directions with the exception of YBant; however, the extent of this effect varies by dance genre. FLA and CHF training may be more effective in improving dynamic balance compared to LAT training. The comparison of balance performance between amateur dancers, rather than professional dancers, and non-participants has significant practical implications for determining whether dance can improve balance in individuals who are not professional dancers, athletes, or who do not regularly participate in sports.

## Supporting information

**S1 File. Data.**
(XLSX)

## Acknowledgments

We sincerely thank all the participants involved in this study, as well as Amor de Dios, Sara Martin Flamenco, Portalo's Baile and other dance schools for their support and collaboration.

## Author contributions

**Conceptualization:** Ningyi Zhang, Sebastián Gómez-Lozano, Ross Armstrong, Hui Liu, Ce Guo, Alfonso Vargas-Macías.

**Data curation:** Ningyi Zhang, Ross Armstrong, Ce Guo.

**Formal analysis:** Ce Guo.

**Investigation:** Sebastián Gómez-Lozano, Alfonso Vargas-Macías.

**Methodology:** Ningyi Zhang, Ross Armstrong, Ce Guo, Alfonso Vargas-Macías.

**Project administration:** Ningyi Zhang, Alfonso Vargas-Macías.

**Resources:** Ningyi Zhang, Hui Liu.

**Software:** Ningyi Zhang, Hui Liu, Ce Guo, Alfonso Vargas-Macías.

**Supervision:** Sebastián Gómez-Lozano, Hui Liu.

**Validation:** Ningyi Zhang, Sebastián Gómez-Lozano, Ross Armstrong, Hui Liu.

**Visualization:** Ningyi Zhang, Sebastián Gómez-Lozano, Ross Armstrong.

**Writing – original draft:** Ningyi Zhang, Ce Guo.

**Writing – review & editing:** Ningyi Zhang, Sebastián Gómez-Lozano, Ross Armstrong, Hui Liu, Ce Guo, Alfonso Vargas-Macías.

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
