## [Decision Letter · Decision Letter 0]

17 Jun 2025

Dear Dr. Ce Guo,

Thank you for submitting your manuscript to PLOS ONE. After careful consideration, we feel that it has merit but does not fully meet PLOS ONE’s publication criteria as it currently stands. Therefore, we invite you to submit a revised version of the manuscript that addresses the points raised during the review process.

We look forward to receiving your revised manuscript.

Kind regards,

Tadashi Ito

Academic Editor

PLOS ONE

Journal Requirements:

Reviewers' comments:

Reviewer's Responses to Questions

**Comments to the Author**

1. Is the manuscript technically sound, and do the data support the conclusions?

Reviewer #1: Partly

Reviewer #2: Yes

2. Has the statistical analysis been performed appropriately and rigorously?

Reviewer #1: Yes

Reviewer #2: N/A

3. Have the authors made all data underlying the findings in their manuscript fully available?

Reviewer #1: Yes

Reviewer #2: Yes

4. Is the manuscript presented in an intelligible fashion and written in standard English?

Reviewer #1: Yes

Reviewer #2: No

Reviewer #1: Dear Authors,

Based on a thorough review of the manuscript "A Comparison of Dynamic Balance Performance Between Non-Dancers and Amateur Dancers Across Three Distinct Dance Genres: A Cross-Sectional Study," the article presents a relevant topic with potential practical implications. However, there are several significant methodological and reporting issues that must be addressed before the manuscript can be considered for publication.

The comments are attached.

Reviewer #2: Thank you for submitting your manuscript. The topic of your study is relevant and well-framed. Comparing dynamic balance performance across amateur dancers of different styles and non-dancers fills a useful gap, especially considering the focus on amateur rather than professional dancers.

Clear rationale for choosing amateur dancers over professionals.

Use of the Y-Balance Test (YBT), a validated and reliable measure for dynamic balance.

Balanced gender representation and clear inclusion/exclusion criteria.

Good discussion linking biomechanical aspects of each dance style to balance outcomes.

While you have calculated p-values and Cohen’s d, some interpretations of significance vs. non-significance are inconsistent. For instance, some findings are labeled statistically significant despite a very small effect size (d < 0.2), which contradicts interpretation norms.

Consider revising statistical interpretations with more caution regarding effect sizes.

Although YBT is a strong test, it only measures reach and not center of mass control or reactive balance. Please acknowledge this as a limitation.

The Latin group had the lowest sample size (n=15), which may limit the power to detect differences. Please mention this limitation in the discussion section.

The manuscript is generally understandable, but many sentences are overly long and would benefit from editing for clarity. A thorough language edit is strongly recommended to improve fluency and scientific tone.

Some parts of the discussion are too descriptive and could benefit from more critical reflection. For instance, possible cultural, biomechanical, or training style reasons behind the non-significance in the anterior direction could be more deeply analyzed.

Define abbreviations like YBcom, DL, NDL, etc., again in figure legends for clarity.

Ensure consistency in capitalization of terms like “control group” or “Latin group” across the manuscript.

Add a clearer rationale in the introduction for choosing those specific three dance genres.

I recommend minor revision. With modest improvements in interpretation, language, and acknowledgement of limitations, this paper would make a meaningful contribution.

**Do you want your identity to be public for this peer review?** For information about this choice, including consent withdrawal, please see our Privacy Policy

Reviewer #1: **Yes:** Rasool Abedanzadeh

Reviewer #2: No

---

## [Author Response · Author response to Decision Letter 1]

20 Sep 2025

All responses have been provided in the attached documents, labelled as “Response to Reviewer 1” and “Response to Reviewer 2”.

---

## [Decision Letter · Decision Letter 1]

23 Dec 2025

A Comparison of Dynamic Balance Performance Between Non-Dancers and Amateur Dancers Across Three Distinct Dance Genres: A Cross-Sectional Study

PONE-D-25-08511R1

Dear Dr. Ce Guo,

We’re pleased to inform you that your manuscript has been judged scientifically suitable for publication and will be formally accepted for publication once it meets all outstanding technical requirements.

Kind regards,

Tadashi Ito

Academic Editor

PLOS One

Additional Editor Comments (optional):

Reviewers' comments:

Reviewer's Responses to Questions

**Comments to the Author**

Reviewer #1: All comments have been addressed

2. Is the manuscript technically sound, and do the data support the conclusions?

Reviewer #1: Yes

3. Has the statistical analysis been performed appropriately and rigorously?

Reviewer #1: Yes

4. Have the authors made all data underlying the findings in their manuscript fully available?

Reviewer #1: Yes

5. Is the manuscript presented in an intelligible fashion and written in standard English?

Reviewer #1: Yes

Reviewer #1: The authors have substantially revised the manuscript and responded thoughtfully to nearly all reviewer comments. The study’s rationale, methodological clarity, statistical reporting, and discussion have all been improved. Remaining minor issues (e.g., lack of a priori power analysis) do not substantially weaken the manuscript’s scientific contribution.

**Do you want your identity to be public for this peer review?** For information about this choice, including consent withdrawal, please see our Privacy Policy

Reviewer #1: **Yes:** Rasool Abedanzdeh

---

## [Editor Report · Acceptance letter]

PONE-D-25-08511R1

PLOS One

Dear Dr. Guo,

I'm pleased to inform you that your manuscript has been deemed suitable for publication in PLOS One. Congratulations! Your manuscript is now being handed over to our production team.

Kind regards,

on behalf of

Dr. Tadashi Ito

Academic Editor

PLOS One